



# Imaging crustal structures through a passive seismic imaging approach in a mining area in Saxony, Germany

Hossein Hassani[1], Felix Hloušek [1], Stefan Buske[1], Olaf Wallner[2]

[1]Institut für Geophysik und Geoinformatik, TU Bergakademie Freiberg, 09599 Freiberg, Germany
[2]Wismut GmbH, Jagdschänkenstraße 29, 09117 Chemnitz, Germany

*Correspondence to*: Hossein Hassani (hossein-hassani@web.de)

**Abstract.** We have used several flooding induced microseismic events that occurred in an abandoned mining area to image geological structures close to the hypocentres in the vicinity of the mine. The events have been located using a migration-based localization approach. We used the recorded full waveforms of these localized microseismic events and have
processed these passive source data as if they resulted from active sources at the known hypocentre location and origin time defined by the applied location approach. The imaging was then performed by using a focusing 3D prestack depth migration approach for the secondary P-wave arrivals. The needed 3D migration velocity model was taken from a recent 3D active (controlled-source) seismic survey in that area. We observed several clear and pronounced reflectors in our obtained 3D seismic image cube, some of them related to a major fault zone in that area and some correlating well with information from
the nearby mining activities. We compared our results to the 3D seismic image cube obtained directly from the 3D active seismic survey and have found new structures with our approach that were not know yet, probably because of their steep dips which the 3D active seismic survey had not illuminated. The location of the hypocentres at depth with respect to the illumination angles of those structures proved to be favourable in that case, and our 3D passive image complements the 3D active seismic image in an elegant way thereby revealing new structures that cannot be imaged otherwise with surface
seismic configurations alone.

## 1 Introduction

Active (controlled-source) seismic surveying is widely used in academia or industry for various purposes, e.g. hydrocarbon exploration, mineral prospecting, geothermal reservoir characterization, or more general studies of the Earth's crust. The methodology for processing of active seismic data sets is very well developed and advanced.

On the contrary, passive seismic imaging (PSI) using natural seismic sources like earthquakes has been subject of research for some time (Soma et al. 2002; Asanuma et al., 2011) but still it is less well established and advanced. Nevertheless, passive seismic imaging has several advantages which makes it in some cases very attractive compared to conventional active seismic imaging: lower costs of data acquisition, because no sources (explosives, vibrators, etc.) are needed; no



environmental impact and no topographical or logistical restrictions with respect to the distribution of sources; usually
greater source energy; etc.

However, PSI methods have their own requirements and limitations: an existing seismic monitoring network with a
sufficiently large number of seismic stations is needed; the position and distribution of sources with respect to target
structures as well the frequency of occurrence and the magnitude of events cannot be controlled; events with larger
magnitude occur on fault planes that act as source which cannot be considered as point sources anymore and may include the
target structures, i.e. the energy source and the reflectors are physically not separated.

A specific characteristic of PSI methods is their ability to image near vertical structures which makes them advantageous
especially in cases where the target structure is characterized by a steeply dipping angle. Reshetnikov et al. (2010) used
microseismic events at the San Andreas Fault system to image near-vertical reflectors in the vicinity of a borehole using the
Fresnel volume migration approach (Lüth et al., 2005; Buske et al., 2009). In comparison to the results of active imaging
surveys in the same area, the near vertical reflectors related to some strands of the fault systems are imaged clearly and with
a significantly improved resolution.

So far, different attempts and methods were employed to produce images of the subsurface using passive seismic sources.
Daneshvar et al. (1995) used direct waves of microearthquakes recorded on the surface to detect shallow structures. In this
method, near-vertical incidence arrivals have been used for imaging discontinuities. The autocorrelation of transmitted
(direct) waves from different sources recorded at individual stations were consistent to the contrast in acoustic impedance of
the shallow structures. Soma et al. (2002) applied a passive seismic reflection technique in which the 3D particle motion
(hodogram) recorded at a seismic station has been analysed to detect reflected waves which are covered within the direct
wave coda. The imaging approach was then conducted using S-wave reflections and their directivity with respect to the
linearity of the particle motion. This technique has been developed and applied for high frequency signals (~200 Hz) and is
advantageous as it is able to image reflectors using a single 3-component geophone or seismometer.  Another PSI method
with a similar concept, is the use of groups of microseismic events which have almost similar waveforms ("microseismic
multiplet") as seismic sources (Asanuma et al., 2011). In this method, using 3-component seismic data, reflections are
detected among the records by analysing 3D hodograms within a coherency function (Asanuma et al., 2001) which measures
the coherency between the recorded wavefield of neighbouring events. The reflected phase is then migrated using a pre-
existing velocity model and a restriction factor calculated based on the P- and S-wave polarization to focus the migrated
amplitudes to the reflection points at depth.

Usually, in PSI applications for imaging the shallow Earth's crust, the sources are either induced or triggered (stimulated)
microseismic events or natural microearthquakes. Stimulated microseismic events are frequent phenomena in mining areas
and hydrocarbon reservoirs (McGarr et al., 2002) due to excavations and the resulting stress changes or simple variations in
pore pressure. In the Schlema-Alberoda mining area, located in south-west Saxony (Germany), mining activities started in
1946 and the mine was in production under the SDAG (Sowjetisch-Deutsche Aktiengesellschaft) Wismut. The mining

operation in this area, targeting a large vein-type uranium deposit (Hiller and Schuppan, 2008), stopped in 1990. During the mining operations up until now, Wismut GmbH (formerly SDAG Wismut) is continuously monitoring the seismicity of the area.

In 2012, an active 3D seismic survey was conducted in the region covering an area of about 10 km x 13 km including the Schlema-Alberoda mining area in order to explore the potential of using pre-existing fault zones as hydraulic paths and natural heat exchanger within a geothermal energy production scenario (Lüschen et al., 2015). Through this survey, several structures were accurately imaged using state of the art imaging techniques (Hloušek et al., 2015). Nevertheless, some geologically expected reflectors did not show up throughout the full resulting 3D seismic image cube, especially a known

major fault (the Roter Kamm) crossing the survey area which is part of the regional Gera-Jàchymov fault zone and can be potentially very useful for exploiting geothermal energy in this area. The Roter Kamm was imaged clearly as a strong reflector in some parts of the 3D seismic cube, while it did not show up as a reflector in other parts where its existence was only concluded from small offsets in reflectors dipping in opposite direction and crossing the Roter Kamm (Hloušek et al., 2015, Schreiter et al., 2015). One reason for this missing direct reflection could be the steep dip (~50-60 degrees) of the

Roter Kamm combined with an insufficient surface coverage of sources and receivers, so that the Roter Kamm is simply not illuminated by the 3D surface reflection seismic survey.

Here in this study, to achieve a better understanding about the crustal structures in the Schlema-Alberoda mining area, especially about the extension of the Roter Kamm, we attempted to image the subsurface through processing and imaging using microseismic events. The existing 3D seismic image obtained from the aforementioned active seismic survey gives us

the unique possibility to compare the results from the passive seismic approach against it and to evaluate the reliability and accuracy of the final PSI results.

## 2 Investigation area geology and seismicity

Hiller and Schuppan (2008) provide a comprehensive overview on the geology of the Schlema-Alberoda mining area. The mine is located in a seismologically active region where the Gera-Jàchymov fault zone intersects the Lössnitz-Zwönitz

syncline (Fig. 1). In this area, the subsurface consists of heterogeneous crystalline rocks and granitic plutons are the most dominant geological features.

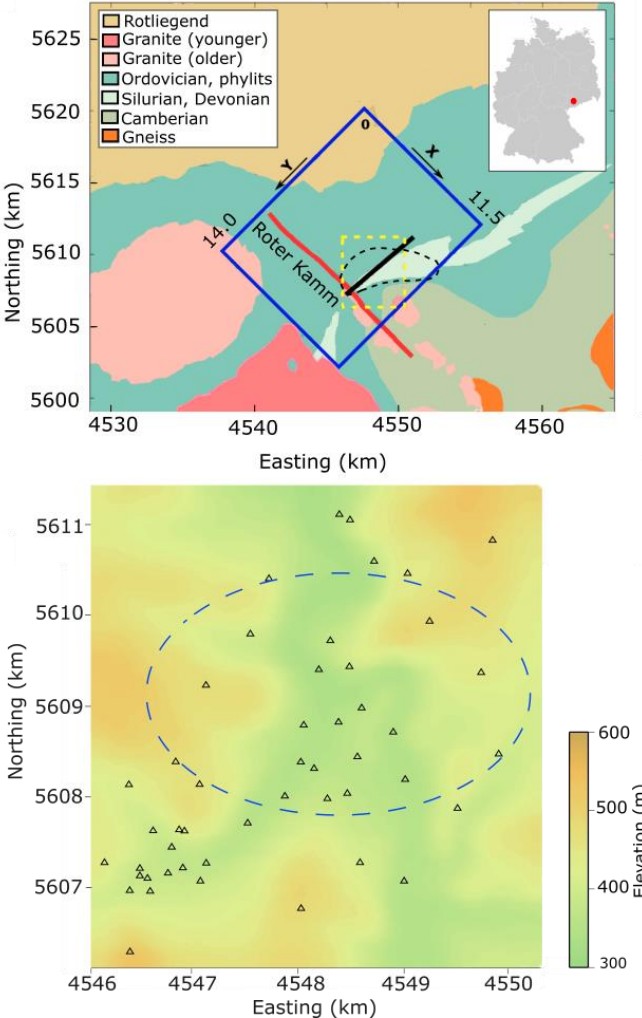

**Figure 1**: Geological map of the study area (top). The blue square shows the position of the 3D model in this study and the 3D active seismic image cube; the yellow dashed square refers to the extent of the local seismic network and the black solid line shows the position of the vertical profile in Fig. 2; the approximate border of the mining area is marked by the black dashed line. The local seismic network used in this study (bottom). The stations are shown by triangles and the blue dashed ellipsoid marks the area where the hypocentres used in this study are located.

The Roter Kamm is a major fault in the area, dipping with an angle of about 50-60 degrees towards the north-east (Fig. 2). The fault zone itself is a vein structure and in some parts caused locally 580 m maximum vertical displacement in the top of granite. The known thickness of the fault zone is between 25-100 m. Different veins such as granite apophyses, aplite dykes and all formations of hydrothermal veins are formed on the Roter Kamm's fault plane. Several other faults with opposite dip towards the south-west but with similar dip angles of 50-70 degrees reach into the granitic body and are conjugated to the





Roter Kamm fault. These faults are ore bearing veins and are mineralized within the Silurian-Devonian schists in the mine and were clearly imaged within the granite by the active seismic survey.

The Schlema-Alberoda uranium deposit itself is located north-east of the Roter Kamm fault (Fig. 1) and is separated from the Schneeberg bismuth-cobalt-silver-uranium deposit by this formation. Figure 2 shows a vertical geological profile through the area with the known and during the mining activities mapped faults.

During the last centuries up until the beginning of the mining operations in 1946, over 100 earthquakes were documented which occurred in this area (Grünthal, 1988; Leydecker, 2007). Besides the natural seismological activity of the area, the excavations in the mine and the subsequent changes in the crustal stress regime likely resulted in more frequently occurring seismic events. Since 1990, after the active mining was abandoned, the mine is subsequently flooded and a seismic network installed over the mining area by the Wismut GmbH (Fig. 1) records microseismic events. Previous studies showed that these events are mostly triggered microearthquakes due to the flooding and the located hypocenters verify the extension of the pre-existing faults into the granitic body (Wallner 2009, Hassani et al., 2018).

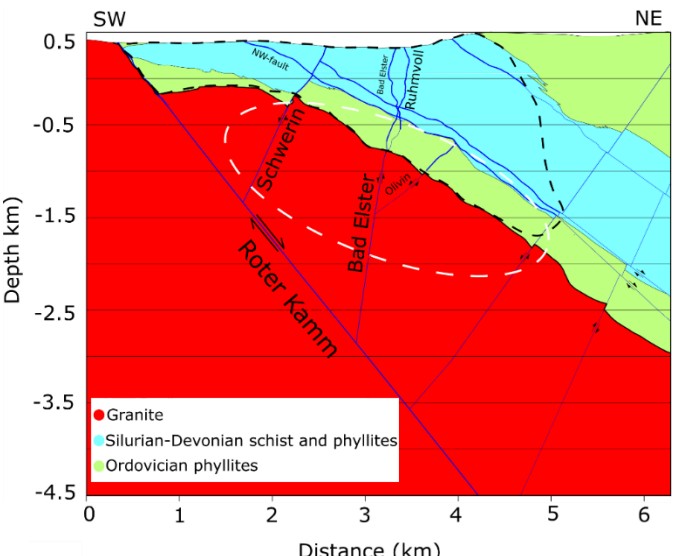

**Figure 2**: Vertical profile of the investigation area developed by Wallner et al. (2009). The black dashed line shows the approximate border of the mining area and the white ellipse shows the area where the hypocentres used in this study are located. For simplification, not all of the known faults are depicted in this figure.

## 3 Principles of 3D Coherency migration

During the last decades, several methods have been developed for migrating seismic data. One of the most known and commonly used methods is Kirchhoff Prestack Depth Migration (KPSDM) which is based on the Kirchhoff integral solution for scalar wave equation (Schneider, 1978):



$$I(m) = \frac{-1}{2\pi} \int_A \int w(m,r) \frac{\partial}{\partial t} u(r, t_s + t_R) dr \tag{1}$$

In this equation, $t_s$ and $t_R$ are the traveltimes from the image point in the subsurface to the source and receiver, respectively. The image value $I$ at each image-point $m(x,y,z)$ in the subsurface is calculated by integrating the recorded wavefield $u$ along the diffraction surface $(t_s + t_R)$. The weighting factor $w(m,r)$ takes the effect of wavefield directivity at the receivers

and the geometrical spreading into account. The time derivative of the recorded wavefield $(\partial u / \partial t)$ accounts for the amplitude correction proportional to frequency and the phase correction which is 90 degrees phase shift in 3D migration.

A disadvantage of KPSDM is that the amplitudes are smeared along the whole two-way traveltime (TWT) isochrones. This can introduce severe migration artefacts, especially in the case of insufficient data coverage and aperture.

An effective solution to reduce wavefield smearing is to take the coherency of the recorded amplitudes into account. Neidell

and Taner (1971) introduced a coherency measurement as a semblance coefficient for a single shot gather which evaluates the coherency between the amplitudes in neighbouring traces. This coherency measurement can be applied as an additional weighting factor within KPDSM to focus the imaged reflected amplitudes onto the real physical diffraction points in depth during the wavefield summation. This approach is called Coherency migration (Hloušek et al., 2015) and is defined as:

$$I(m) = \frac{-1}{2\pi} \int_A \int C_S(m,r) w(m,r) \frac{\partial}{\partial t} u(r, t_s + t_R) dr \tag{2}$$

The term $C_s(m,r)$ is the weighting function based on the semblance coefficient:

$$C_S(m,r) = \frac{\int_{-T/2}^{T/2} \left| \sum_{i=1}^{N} u_i \left( t + t_s + t_{r_i} \right) \right|^2 dt}{N \int_{-T/2}^{T/2} \sum_{i=1}^{N} \left| u_i \left( t + t_s + t_{r_i} \right) \right|^2 dt} \tag{3}$$

This function is defined for each image point $m$ and each receiver $r$ in a shot gather and represents the coherent energy of a wavefield relatively to its total energy within a defined time window $T$ and over $N$ neighbouring traces. The length of the time window should be chosen according to the dominant frequency of the source wavelet. This ratio varies between 0 for no

coherency (e.g. random noise) and 1 for a perfectly coherent wavefield. Thus, the smearing of the amplitudes is focused to the physically contributing part along the migration operator (TWT isochrone), i.e. the diffraction or reflection point.

The principle of Coherency migration is shown in Fig. 3. Suppose that the diffraction point D lying in a constant velocity medium is to be imaged using a source located at depth and an array of receivers on the surface (star and triangles in Fig. 3, respectively). If the source releases a single wavelet (Fig. 3-a), the resulting wavefield diffracted from D and recorded by the

receiver array at the surface will be as shown in Fig. 3-b. The dashed ellipsoid in Fig. 3-a is the TWT isochrone $(t_s + t_R)$ corresponding to one receiver (blue). Migrating the recorded wavefield of this receiver through KPSDM smears the amplitudes along the whole TWT isochrone (Fig. 3-e) and all of the points on the isochrone can be considered as potential


diffraction points. By applying coherency migration to this single trace using 10 neighbouring traces for the weighting factor

Cs (5 traces on each side), the migrated signal gains a high coherency factor around point D during migration (Fig. 3-f). This

is because the recorded amplitudes at neighbouring receivers are coherent regarding to the diffraction point D, i.e. they

follow the calculated traveltime pattern of point D within a defined time window $T$ (marked in red in Fig. 3-b). In other

words, we measure the semblance of the wavefield at neighbouring traces within the time window. On the other hand, if we

consider a hypothetical diffraction point D' on the TWT isochrone (Fig. 3-c), the recorded amplitudes at the neighbouring

traces do not follow the calculated traveltime pattern of this point (Fig. 3-d), thus the coherency factor for this point is low

and the imaged amplitudes will be decreased around it during migration (Fig. 3-f).

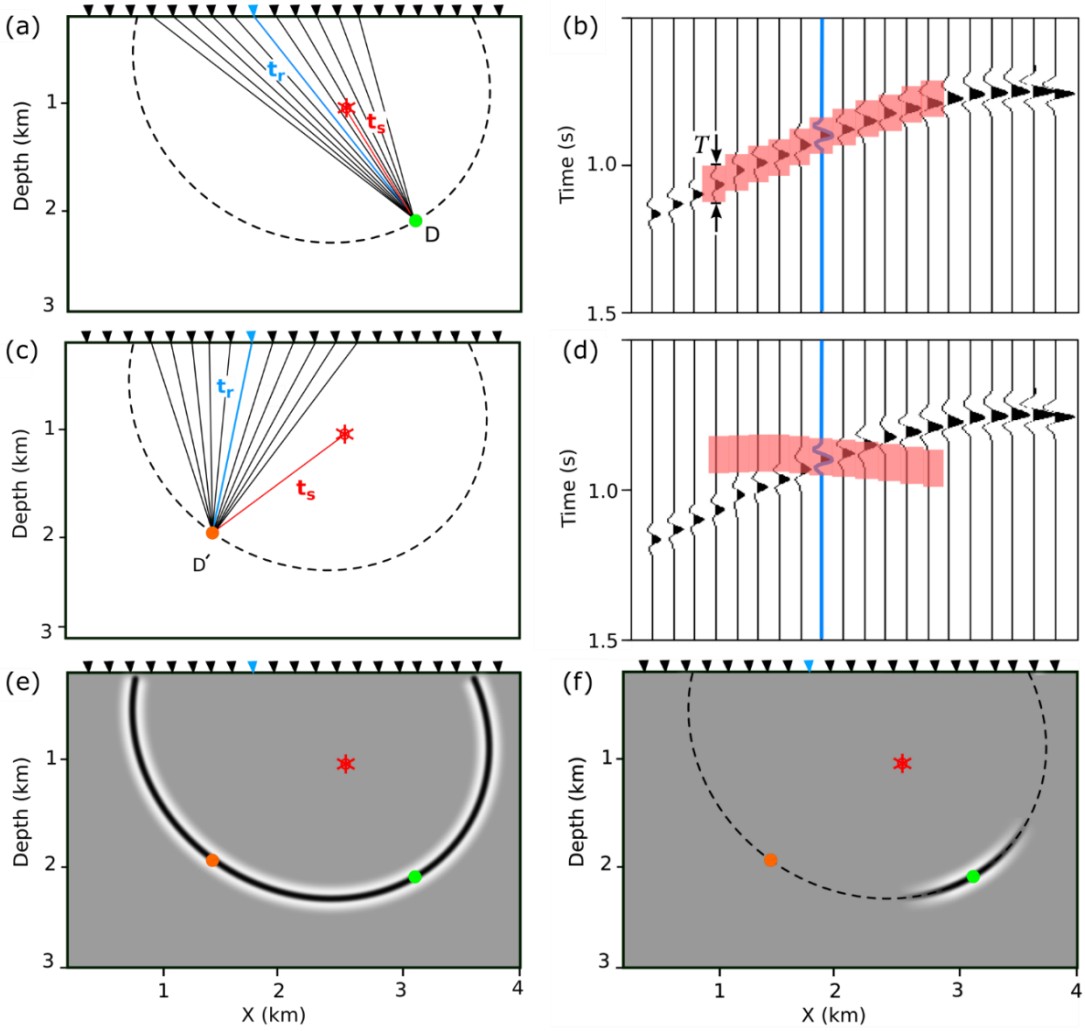

**Figure 3:** Concept of coherency migration. The red asterisk shows the source position and the triangles are surface receivers. The red boxes in (b) and (d) are the time windows with the length T defined on the neighbouring traces, with respect to the traveltime of the image point. For detailed description see text.



## 4 Imaging procedure

### 4.1 Locating microseismic events

In contrast to active seismic surveys, passive seismic imaging requires an additional step, because the location of the sources is initially unknown. Since the accuracy of every seismic imaging survey is highly dependent on the accurate knowledge of the source-receiver geometry, an initial and important step in PSI is to precisely locate the hypocenters.

In this study, we used 136 microseismic events ($-1.30 < M_w < 0.90$) in the Schlema-Alberoda mining area which occurred between 1998 and 2012 and which were recorded by the local seismic network operated by Wismut GmbH. The network (Fig. 1) consists of 56 stations from which 54 stations are equipped with 4.5 Hz 1-component (vertical) geophones and 2 stations are downhole hydrophones. To accurately locate the events with a minimal error in location and origin time, we applied a migration-based earthquake localisation algorithm (Hassani et al., 2018) using only P-wave arrival times. The hypocenters are located with an uncertainty of approximately 50 m in space and 5 ms in source (origin) time, respectively.

### 4.2 Data analysis

Generally in PSI, not all available data might be appropriate to be used in the imaging procedure. Although we have located the sources with a reliable accuracy, only those source-receiver pairs that have maximum S/N ratio were selected for the imaging procedure. The records of the two borehole hydrophones were excluded from the imaging procedure because of their insufficient signal-to-noise ratio.

Since our dataset contains only vertical component records, it was more meaningful to rely only on P-wave reflections for imaging. Thus we selected only those records with clear and strong (in comparison to maximum amplitude on the trace) direct P phases. This implies that the source released enough energy in the form of P-waves such that the P-wave reflection amplitudes can be clearly detected on the records within other phases. These considerations led to a selection of a total of 170 records from 84 microseismic events.

### 4.3 Migration

Basically, the coherency migration is performed for individual source gathers because the source signal must be common between the neighbouring traces over which the coherency value is to be calculated (Neidell and Taner, 1971). The presumption is that all the receivers are identical in their physical properties (e.g. natural frequency) and they have similar ground coupling conditions such that the recorded traces have similar characteristics (frequency content and phases). This condition is valid for most active seismic surveys. Normally the distance between the geophones does not exceed some tens of meters and the ground over which the neighbouring geophones are installed has comparable properties in the sense of wave propagation and there is no significant difference in the frequency content of the recorded wavefield at neighbouring geophones.



The recording network used here in this study has not been primarily designed for a reflection seismic survey. The function of this network is only to detect seismic events and therefore the waveforms were not of primary importance in its design. The receivers are attached to different grounds like solid rock, concrete based surface, weathering surface, etc. On the other hand, due to the large distances between the neighbouring stations (up to several hundred meters), the characteristics of the underlying layers may vary significantly from one station to another. This causes differences in frequency content of the recorded signal between the stations for a single event. Moreover, the source mechanism of an event with respect to the position of the receivers can cause different waveforms at different stations. Figure 4 (a and b) shows examples of the selected records of two representative events (source gathers). As can be seen, the wavefield for the same event appears with different frequency content at different stations. Comparing the recorded traces between all source gathers, we were able to identify a similarity in the frequency content as well as in the waveform between the records of different events at the same individual receiver (Fig. 4-c). This makes it possible to apply the Coherency migration to common-receiver gathers (CRGs).

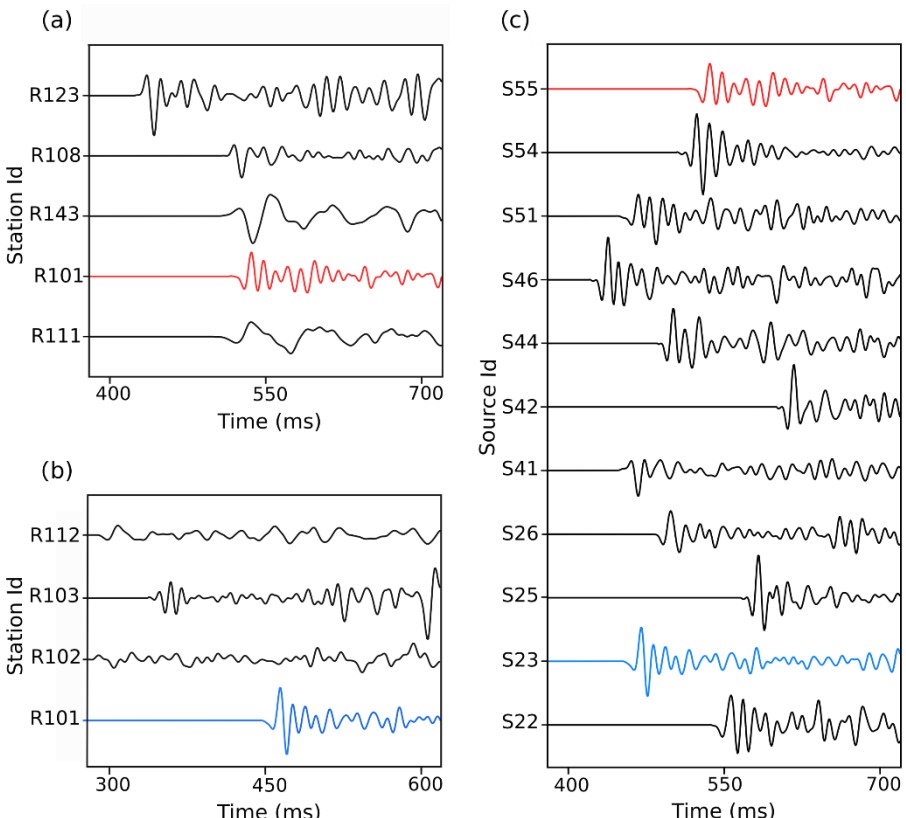

**Figure 4:** A comparison between common source gathers S55 and S23 (a and b) (same events, different receivers) and a common receiver gather R101 (c) (same receiver, different events). R101 is marked in red and blue for events S55 and S23, respectively.

The 170 traces which were selected for the migration procedure are grouped into 30 CRGs. Moreover, some of the CRGs contain very few traces and in order to achieve a reliable coherency measure (see Eq. 3), these CRGs are excluded from the





migration. Finally 10 CRGs including records from 48 events with a total number of 128 traces were chosen for migration.
The CRGs with the minimum and maximum number of traces include 7 and 26 sources, respectively. During migration, all traces in each CRG are involved in calculating coherency factor as neighbouring traces. After evaluating the direct P-wave wavelength in all traces involved in migration procedure, each CRG is assigned an individual length $T$ (varying between 16 and 42ms) for the time window over which the coherency value must be calculated (Eq. 3). To eliminate the effect of varying focal mechanism on the polarity of the recorded wavefields, the recorded traces are adjusted to the same polarity.

To magnify the focusing effect of the Coherency migration, an exponent $\propto$ can be applied to the coherency function in Eq. (2) as $C_S^\propto(m,r)$. A higher value of $\propto$ intensifies the contrast between the most coherent amplitudes and the less coherent ones and random noise. Nevertheless, to avoid exaggerating the coherent signals and the resulting ghost reflectors in the final image by choosing a too high $\propto$ value, different values must be tested to find the optimum one. In this study we used an exponent value of 3 in the coherency migration.

For calculating traveltimes, we applied a finite difference approximation of the eikonal equation proposed by Podvin and Lecomte (1991) and used a 3D P-wave velocity model of the area derived from first-arrival traveltime tomography of the 3D active seismic data set (Hloušek et al., 2015). The same velocity model was used for imaging the 3D active seismic data set as well as to locate the hypocenters (Hassani et al., 2018). Since the traveltimes are calculated based on the P-wave velocity, it is expected that the S-wave reflections gain a low coherency factor and through the exponent value ($\propto = 3$), the reflected
S-wave amplitudes are suppressed during the coherency migration. This should assure that only P-wave reflections are the dominant constructive migrated amplitudes in the final image.

## 5 Imaging results and analysis

To obtain the final image, different methods can be applied, such as migrating envelopes, absolute values or original (phase-consistent) values of the wavefield. In our study, we used the absolute and original values for imaging. The coherency
migration is applied to the CRGs, resulting in an individual 3D seismic image cube for each single CRG. The final image is then obtained by stacking these single images. Figure 5 shows the imaging procedure.

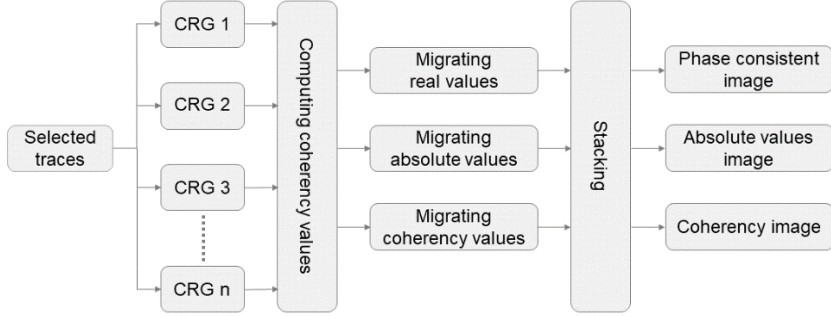

**Figure 5**: Schematic sketch of the imaging procedure.



To get noticeable image values from low-amplitude coherent signals and to eliminate the effect of amplitude distortions due to the presence of the S-wave and its reflections, the coherency value calculated during migration is also considered as an

image value. The resulting image is called "coherency image" in the following.

**Figure 6:** Top and side views of the 3D model, the position of sources (red dots) and receivers (blue triangles) that contributed in migration procedure and the major geological structures (top and middle-left). A slice through the coherency image (middle-right), the phase-consistent image (bottom-left) and the absolute values image (bottom-right). The dashed line ABC refers to the position of the shown image slices.



Figure 6 shows the position of the sources and receivers used in the migration procedure and a vertical slice through the three resulting seismic image cubes. As can be seen, the sources and receivers have a very limited coverage over the image cube. Therefore we focus our analysis on the parts of the image in the vicinity of the sources and receivers. For comparison and further analysis, the size of the 3D image cube is defined the same as the 3D seismic image cube obtained from the active seismic survey in the area (Hloušek et al., 2015).

Comparing the three images in Fig. 6, it is clear that the coherency image shows the clearest and most pronounced image of the structures. Particularly, in deeper parts (>4km) we can detect some reflectors which are not clearly visible in the two other images. However, a dominant reflector is clearly visible in all of the three images in shallower parts close to the position of the sources (red line in Fig. 7). This structure will be discussed in following.

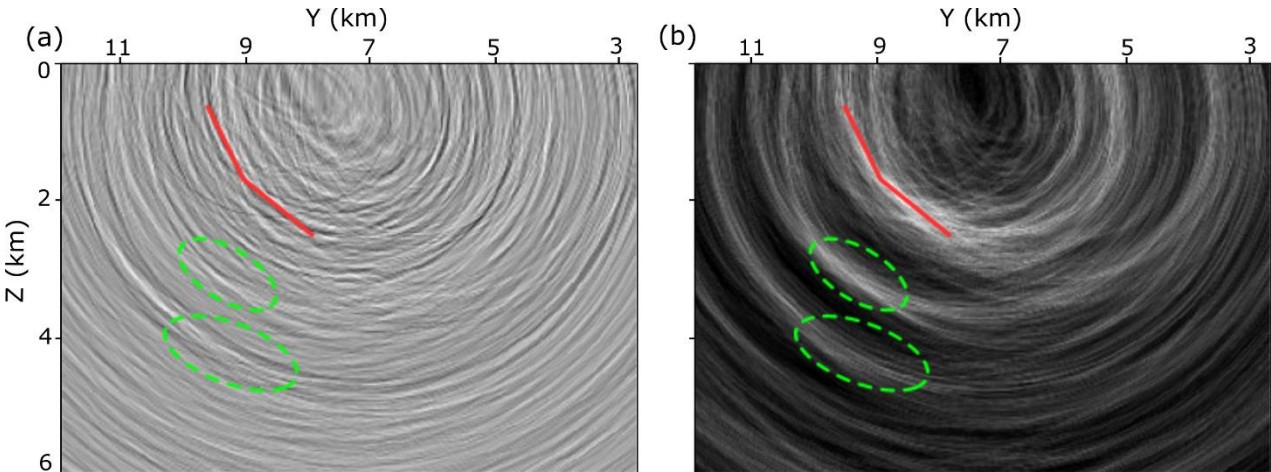

**Figure 7:** Zoomed-in illustration of the vertical slices through the phase-consistent image (a) and absolute values image (b) shown in Fig. 6. The red solid line shows the imaged part of the Roter Kamm fault and the green dashed ellipses refer to the reflectors B and D in Fig. 11 and 12.

The presence of S-waves with a comparable amplitude to the P-wave is a complication which affects the quality of the final image when we image the amplitudes. Although because of the P-wave velocity model the S-wave phases should not be

stacked constructively during migration, their amplitudes are still present in the final image and somehow distort the imaged P-wave reflections. Besides that, in some of the events, the S-wave has a lower frequency (Fig. 8) which causes less energy lost in its reflections from deeper parts in comparison to those of P-wave. Therefore, reflected P-waves from deeper parts may be covered by stronger S-wave reflections. Rather than very few traces (as the examples in Fig. 8) the S-wave was not clearly detectable in the recorded traces and it was not possible to distinguish the S-waves from the P-waves or even to

separate them and to use them in the imaging procedure.





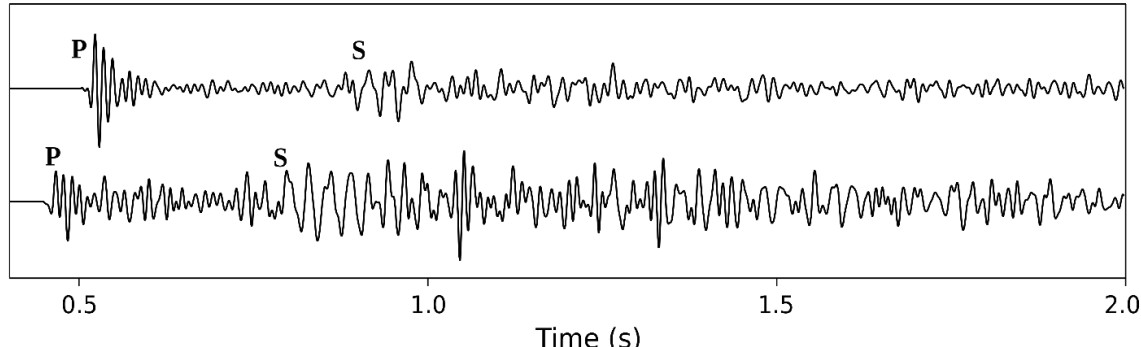

**Figure 8:** An example of the recorded waveforms used in the migration procedure. The P- and S- direct waves are marked for comparing the frequency content.

Nevertheless, because of the P-wave velocity model and since the coherency value is calculated over a time window which represents the P-wavelength, the calculated coherency values are less affected by S-wave and its reflections. Therefore, the

final image would be less distorted when the amplitudes are not directly included in the image. On the other hand, all sources have the same contribution in the coherency image irrespective of their P-wave amplitude. Thus, the coherency image is likely the most reliable one and we focus our analysis on this image. Figure 9 shows sequential vertical slices through the 3D coherency image.

As described in Sect. 2, the major structure in this area is the Roter Kamm fault. The part of this fault observed through

geological surveys during the mining operations reaches a depth of ~300m below sea level. Geological interpretations expect the fault plane to extend linearly down to a depth of 8km. Our final results clearly detect this fault as a prominent reflector which is visible in the coherency image as well as the phase-consistent and absolute values images (Fig. 7 and 10). Nevertheless, the results reveal details about the Roter Kamm's extension which differ from the previous assumptions: the imaged fault plane has a slightly smaller dip angle, it is bending at a depth of ~1400m and extends further towards the north-

east (y direction in our local coordinate) with a smaller dip angle down to ~2600m below sea level (Fig. 10). Due to the limited amount of sources (hypocentres) and receivers (stations at the surface), the extent of the reflector down to greater depths could not be imaged.

The Schwerin fault is another structure which was mapped during mining operations in the area. Surprisingly it can be seen that this fault extends directly towards the bending point of the Roter Kamm (Fig. 10). It is convincing that it reaches the

Roter Kamm at its bending point and this may explain the sudden change in the Roter Kamm's extension direction. For that same reason as described above, the limited source-receiver coverage and illumination, it was not possible to image the Schwerin fault assuming that the fault plane is reflective.



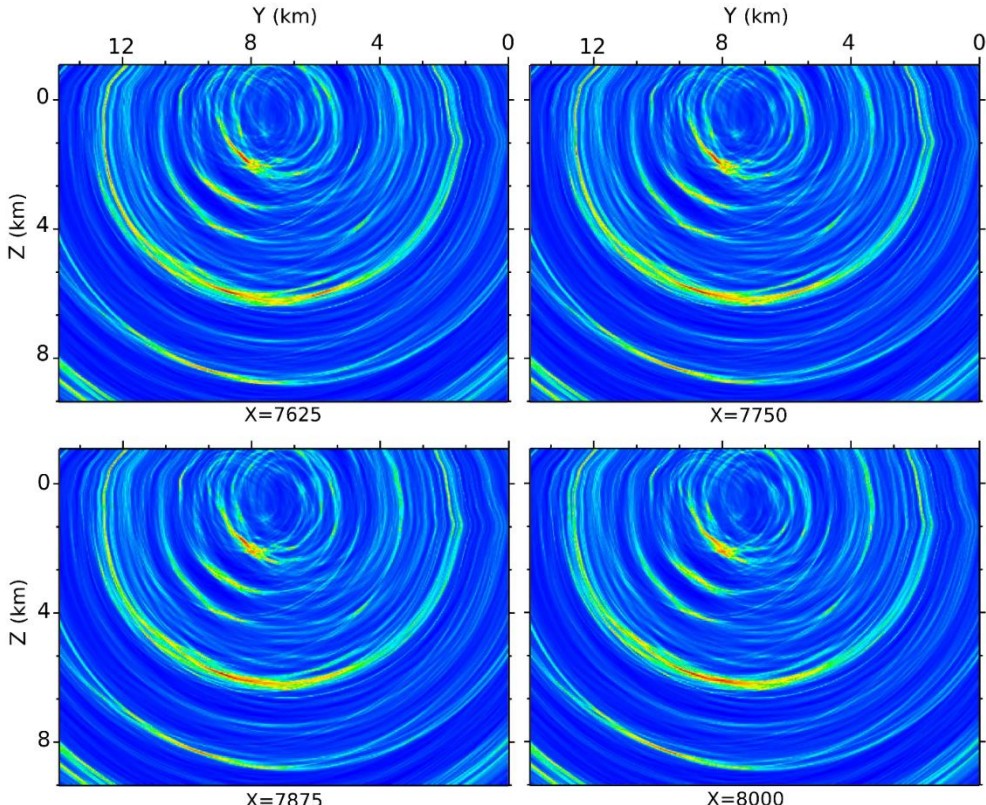

**Figure 9:** Sequential slices over 375 m interval in x direction through the coherency image cube.

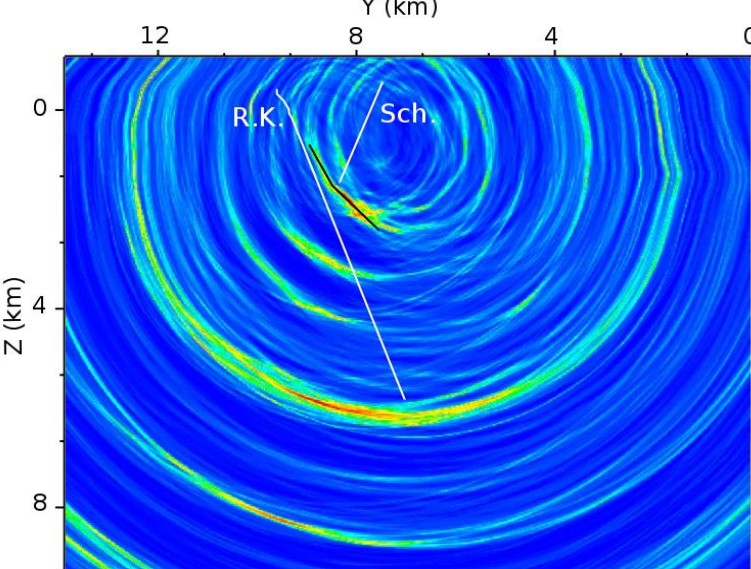

**Figure 10:** Comparison of the coherency image with the presumable Roter Kamm fault (R.K.) and Schwerin fault (Sch.) as derived from the geology map of the area. The deeper parts of these two structures are only extrapolated and not known.





Generally, in an active surface seismic reflection survey, the steeply dipping structures are unlikely to be imaged unless the data contains records of far-offset sources and receivers. Moreover, the target structure must have a high acoustic impedance contrast (reflectivity) so that the body waves from far-offset sources can be reflected effectively with a minimum refraction

at the reflector's boundary. However, this is not a limitation in passive seismic imaging because the sources usually release more energy and the corresponding energy loss is less since they are located in the subsurface and close to the imaging target. Thus, if the receivers are installed in a favourable position, near vertical structures can also be imaged even if they are not strongly reflective. The Roter Kamm is a vein structure and according to its geological properties, it is not in all parts expected to be strongly reflective (see Hiller and Schuppan, 2008; Hloušek et al., 2015; Schreiter et al. 2015). In addition,

this fault has a large dip angle (Fig. 6). Therefore, the fault plane could not be imaged clearly in the active seismic survey in those parts that are considered here in our PSI study.

Nevertheless, Hloušek et al. (2015) show evidences in some parts of the 3D active seismic image which demonstrate the existence of the Roter Kamm, e.g. discontinuities in some other reflectors conjugate to the Roter Kamm (at a distance of ~1 km to the analysis area of this study in x-direction). Also, by stacking the shot gathers with a far-offset to the Roter Kamm,

in a part of the image cube they could detect a clear and strong reflector at the position where the Roter Kamm is expected to be located.

Rather than the Roter Kamm, several other structures are detected in our final image. Figure 11 shows the same vertical slices of the PSI 3D coherency image and the 3D absolute values image resulting from the active seismic survey. Most of the reflectors detected in our final image are also visible in the active seismic image cube (Fig. 11). One of the structures

detected in both images is the so called "Schneeberg Body" (SB) which is a rather diffusive but still strongly reflective zone at a depth of 4-7 km. Its existence was detected first by the aforementioned active seismic survey and Hloušek et al. (2015) give a hypothesis about the nature of the SB and describe it as a highly reflective complex zone. This structure shows higher reflectivity at its top and bottom in the active image whereas in the PSI results the top and bottom of the SB are clearly visible (lines D and F in Fig. 11). Especially at the bottom of the SB, the less reflective dipping tail in the active image

follows the trend of the detected reflector (F) in the passive image.

Nevertheless, the SB does not appear as a diffusive reflective zone in the passive image. In comparison to the active seismic survey, because of the narrow receiver aperture in the PSI survey, the diffusive energy has less contribution in the total reflected energy recorded by all receivers and does not appear in the final image.

Moreover, both images detect reflectors C and E with a rather low reflectivity. These reflectors belong to a group of so

called "conjugate faults" that extend towards the possible deeper extension of the Roter Kamm. Directly above the SB, a fairly strong and well focused reflection can be seen in the coherency image (line B in Fig. 11 and 12), while in the active seismic image, a low reflective structure oriented perpendicular to the reflector B is imaged at the same position. Figure 12 shows a zoomed-in version for this reflector as well as a comparison to the phase-consistent active seismic image. As it can be seen, reflector B appears perpendicularly to the small-scale reflections detected in the active image. This reflector may be





interpreted as a small zone of mineralization related to the SB, nevertheless it is not connected to the uppermost layers. In
       Fig. 12, reflector D also shows a very good correlation to the detected reflection at the top of SB in the active seismic image.
       Reflectors B and D (top of SB) are also visible in the absolute values and phase-consistent images (Fig. 7).

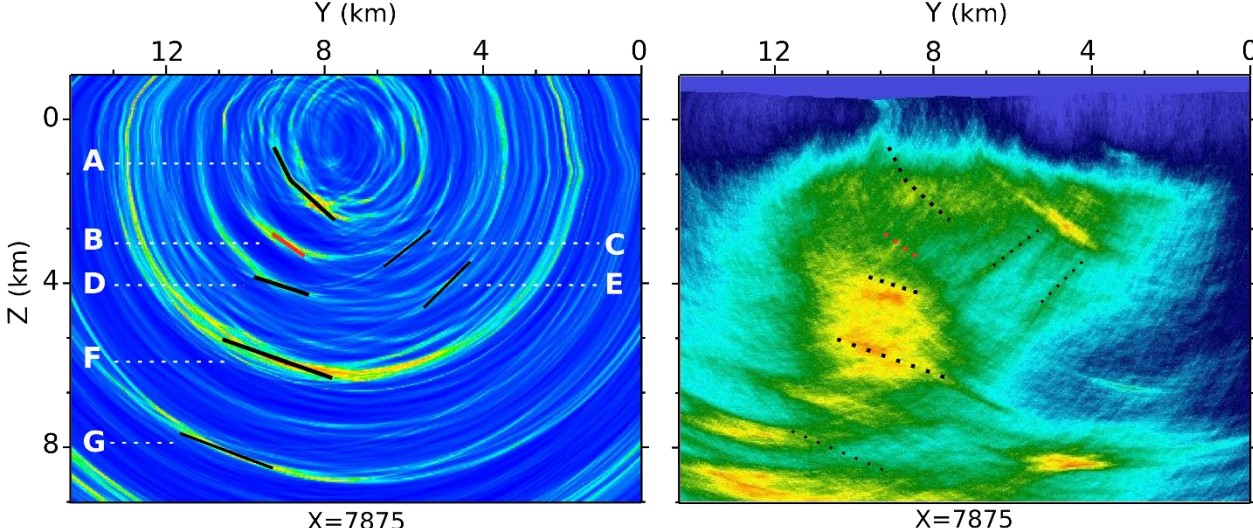

**Figure 11:** Comparison of the imaged structures in the coherency image (left) to the results of the active seismic survey in the area (right).
The thick solid lines show the reflectors with higher image value, the thin solid lines mark the reflectors with a lower image value and the
black dashed lines show them projected onto the active seismic image.

       In the deeper parts of the seismic image cube, another reflector (G) is detected in the coherency image. Comparing it to the
       active image, this reflector has a displacement in the Y direction. It must be noticed that in our dataset, the recorded traces
       were not of the same length in time domain and not all of them allow to image the deeper part of the cube. Therefore, the
       resulting passive seismic image loses resolution at depths greater than 7km. On the other hand, this reflector has a horizontal
distance of ~5km to the position of the sources and receivers illuminating it, which can also cause bias in the position and dip
       of the imaged reflector. Therefore, the detected position of this reflector can be considered to be more reliable in the active
       seismic image.



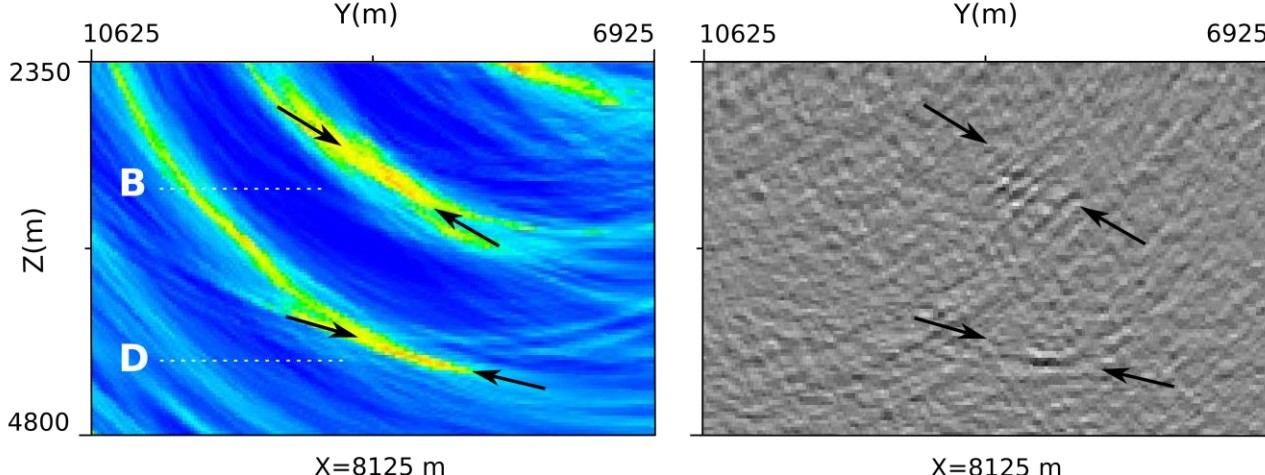

**Figure 12:** A zoomed illustration of the reflectors B and D (left) and their position shown by the arrows on the phase consistent active seismic image (right). Note that these slices are located at a distance of 250 m in x direction to the slices in Fig. 11.

## 6 Conclusions

We have presented a passive seismic imaging approach using the Coherency migration technique and the records of single component (vertical) geophones for imaging crustal structures. Despite the narrow aperture of the sources and receivers over the study area, the imaged structural inventory derived from the results is remarkable. This study showed that the coherency of recorded wavefields can be used directly as an image value to illuminate subsurface structures. This is an advantage of the applied migration method especially for PSI where the sources often have different magnitudes.

The data used in this study was recorded by a permanent local seismic network. This network was not designed for reflection seismic surveys and therefore the dataset was not optimal for conducting a passive seismic imaging survey. In spite of the low number of traces which were appropriate for imaging the structures, our passive imaging approach could yield reliable results. Nevertheless, setting up stations under equal installation conditions and with a wider aperture would further strengthen the results and may be considered for any future passive seismic survey in this area.

The similarity of the recorded waveforms from different sources shows a resemblance between the microseismic sources. This is proof of previous findings about the nature of the current seismicity in the area (Hassani et al., 2018). It was interpreted that the microseismic events beneath the Schlema-Alberoda uranium mine have a common nature and are a sign of crack growth in the granitic body due to the increasing pore pressure and unstable mechanical state of the structures.

In this study, we were able to detect the extension of the Roter Kamm fault towards greater depth despite its steep dip and its assumed low reflectivity. The observed reflections from this fault within the granitic basement proves that a significant

impedance contrast must be present at the fault zone reflector. The resulting images can be used for further studies related to this feature.

Our results show a very good correlation to the results of the previously conducted active seismic survey. In comparison, the advantages of PSI led to a better understanding of some structures with low reflectivity such as the Roter Kamm fault and the Schwerin fault. Furthermore, the correlation between the results of the both passive and active imaging surveys demonstrates their reliability.

The velocity model used in this study does not account for possible anisotropic properties of the underlying medium, but its
reliability is confirmed through tests by Hloušek et al. (2015). Although we do not except a significant error in the results, an anisotropic velocity model could further improve the final image.

**Data availability**

The data is in possession of Wismut GmbH and could be available on request.

**Competing interests**

The authors declare that they have no conflict of interest.

**Acknowledgements**

We would like to thank Wismut GmbH and Mr. Thomas Ebert (C&E GmbH) for their support and providing the dataset for this study.

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
