# Peer review of "Imaging crustal structures through a passive seismic imaging approach in a mining area in Saxony, Germany"

_Solid Earth, 2021_

## Referee Comment (RC2)

[referee-annotated manuscript omitted]

---

## Author Response (AR1)

**Dear Reviewers, Dear Editor,**

We are Thankful for the constructive reviews of our manuscript. Following the reviewers comments, we tried to revise the manuscript for more clarity and to include more information in order to provide a better understanding of our paper for readers.

**Comments (C) and Replies (R):**

Line 1: *"Imaging crustal structures"*

C: In my opinion, "Imaging shallow crustal structures" is more appropriate as the features are restricted to just few kilometres from surface!

R: The presented imaging procedure in this paper is not designed to image only shallow structures and it can also be used to image deep structures. Therefore we preferred to not restrict the title to only shallow crustal structures. The declaration is added to conclusions:

Line 337-349:

-> It should be noticed that the presented imaging procedure can also be applied to image deep crustal structures. To which depth extent can the imaging results be reliable, depends to the magnitude level of the passive sources as well as the spreading of the sources and recording stations.
* * *
Line 10: *"passive source data as if they resulted from active sources"*

C: In order to validate the claim, please provide a comparison panel consisting of a shot gather from active seismic survey, raw & processed common receiver/shot gather later in the article. It does not necessarily have to be from the same location. It will be quiet interesting to see if we can correlate any reflection events between active and passive data. Also, it will be highly appreciable if you put the processing steps involved in handling the passive data in a tabular form.

R: In this sentence we are explaining briefly the imaging method used in this study and not claiming any similarity between the passive and active data (... processed these passive source data as if they resulted from active sources...). The sources used in this study have nothing in common with the active seismic survey discussed later in the article.

A direct comparison between the passive and active seismic surveys is shown in figures 12 and 13 where the correlation between detected reflection events is demonstrated. Also in section 5 (line 288-292) we referenced to the previously published paper about the active seismic survey where the result of stacking common shot gathers is shown and can be compared to our results (especially about reflection events related to the Roter Kamm fault).

A schematic of the imaging procedure steps is shown in Fig. 6.

Line 12: *"secondary P-wave arrivals"*

C: Please cross-check and use the same analogy about which P-wave arrivals were used. Later in section 4.2 Data analysis, it is said that direct (transmitted) P phases were used. I got a little bit puzzled if we are talking about the same phenomenon.

R: As it is described here and later in section 4.2 (line 167), the structures are imaged using the secondary P-wave arrivals (reflections). In section 4.2 where the primary (direct) P-wave arrivals are mentioned, it is described that the observed direct P-wave amplitudes are considered [only] for the selection of the traces to be used in the imaging procedure.
To avoid any misunderstanding, the paragraph (lines 167-171) is modified:

Line 167-171:

Since our dataset contains only vertical component records, it was more meaningful to rely only on P-wave reflections for imaging. To assure that the P-wave reflection amplitudes are large enough to overcome other phases, we selected only those records with clear and strong (in comparison to maximum amplitude on the trace) direct P phases  which implies that the source released enough energy in the form of P-wave . These considerations led to a selection  of 170 records from 84 microseismic events. The dominant frequency bandwidth of the selected traces is 10-150 Hz.

Line 14: *"some correlating well with information from the nearby mining activities"*

C: In my opinion, this has not been stated later in the article which features are these.

R: Here we mean the Roter Kamm fault and the conjugate faults. In section 2 it is mentioned that some information about these features had been released based on the geological map of the area (lines 83, 92, 95-97 and 259-261). Later in section 5, rather than discussing about the Rotar Kamm fault, we mention also the reflectors C and E (Fig. 12) which belong to the conjugate faults group (lines 306-307).

Line 19: *"in an elegant way"*

C: Please be more specific, what do you mean by this!

R: We want to point out the simplicity and cleverness of the method. Reference:
https://www.oxfordlearnersdictionaries.com/definition/english/elegant?q=elegant

Line 29: *"no topographical"*

C: Topology still could be a major problem, may be due to civil restrictions, swapy areas, rapid change in elevations over short distances etc.

R: We described here at the end of the sentence that "no topological or logistical restrictions" is about the seismic sources (and not receivers). Of course passive (natural) seismic sources have no restrictions due to civil restrictions, rapid changes in elevations, etc.
* * *
Line 32: *"the position and distribution of sources with respect to target…"*

C: Kindly break it in smaller sentences, quiet hard to follow!

Thanks for mentioning, the sentence is now braked down to two sentences:

Line 32-33:

-> the position and distribution of sources with respect to target structures cannot be arranged; the frequency of occurrence and the magnitude of events cannot be controlled
* * *
Line 35: *"the energy source and the reflectors are physically not separated"*

C: Is there any criteria followed for the source selection in this case? For example those events originating within the mine area were only taken, events related to faults were rejected etc. As in Figure 2 (white circle), the obvious source for induced seismicity appears to be fault based but majority could be due to increase in pore-pressure.

R: We didn't exclude any sources based on their origin. Nevertheless, referring to the previous publication (Hassani et al., 2018) the events are occurred due the increased pore pressure on the pre-existing faults and are located in the lowermost and (most of them) below the excavated area of the mine.
* * *
Line 39: *"Fresnel volume migration"*

C: Was Fresnel Volume migration also been tested here? If yes, did we obtain any similar results compared to Coherency migration? If we didn't, what might be the possible cause in your opinion?

R: The Fresnel volume migration was not tested in our study.
* * *
Line 52: *"amplitudes to the reflection points at depth"*

C: In my opinion, PSI technique shouldn't be discussed in such great details because we are not attempting any such method here. Rather some more emphasis can be put on CM.

R: We tried to provide a comprehensive overview on some previous attempts of passive seismic imaging. However, this paragraph is modified to exclude some details:

Lines 43-53:

So far, different attempts and methods were employed to produce images of the subsurface using passive seismic sources. Daneshvar et al. (1995) used direct waves of microearthquakes recorded on the surface to detect shallow structures.  between the sources and receivers. The autocorrelation of near-vertical incidence direct waves from different sources recorded at individual receivers showed consistency to the contrast in acoustic impedance of the shallow structures. Soma et al. (2002) applied a passive seismic reflection technique in which the 3D particle motion (hodogram) recorded at a seismic station has been analysed to detect reflected waves which are covered within the direct wave coda.  This technique has been developed and applied for high frequency signals (~200 Hz) and is advantageous as it is able to image reflectors using a single 3-component geophone or seismometer. Another PSI method with a similar concept, is the use of groups of microseismic events which have almost similar waveforms ("microseismic multiplet") as seismic sources (Asanuma et al., 2011). In this method, using 3-component seismic data, reflections are detected among the records by analysing 3D hodograms within a coherency function (Asanuma et al., 2001) which measures the coherency between the recorded wavefield of neighbouring events.
* * *
Line 79: *"Investigation area geology and seismicity"*

C: May be "Geology and seismicity of area"

The title of the section is changed:
Line 79:
-> Geology and seismicity of the area
* * *
Line 83: *"crystalline rocks and granitic plutons are the most"*
C:  as

R: Here we mean that the "granitic plutons" are the most dominant features among other crystalline rocks.
* * *
Figure 2:
C: Adding a line might be helpful!

The Lines are added to Figure 2.
* * *
Line 87: *"The local seismic network"*

C: There were almost no stations in the NW, few in the NE and SW direction. Whereas the density is higher in the SW direction. Did you observe it might have resulted in higher semblance coefficient values because of more receivers over short distances?

R: Measuring the semblance coefficients is performed over every single receiver gather separately (section 4.3), therefore the distance between receivers doesn't have any influence on the calculated values of semblance.
* * *
Line 131: *"dominant frequency of the source wavelet"*

C: As the events are due to flooding induced seismicity, therefore passive data should be composed of variable frequency for each source. Is it possible to showcase the average source wavelet representing the source mechanism?

R: We observed similar frequency content of the recorded traces from different sources (section 4.3; Fig. 4) which proves similarity in the mechanism of the sources. Nevertheless, since the aim of this study was only to image subsurface structures using passive data, we didn't perform any focal mechanism analysis.
* * *
Line 164: *"that have maximum S/N ratio"*

C: Was something like spectral equalisation or amplitude normalisation being tested? It might be helpful in boosting the low frequency signals for weaker events and balanced amplitude traces might result in more traces that could be used for migration.

R: Yes, we did normalized the traces to amplify weaker events, but because of the very small magnitude of the events and the presence of the ambient noise, this process couldn't result in more traces that can be used for migration.
* * *
Line 169: *"direct P phases"*

C: Secondary P (abstract part) or direct P?

R: We modified to avoid any misunderstanding. Explanation as the answer to the previous comment (in abstract):
Lines 167-171:

Since our dataset contains only vertical component records, it was more meaningful to rely only on P-wave reflections for imaging. To assure that the P-wave reflection amplitudes are large enough to overcome other phases, we selected only those records with clear and strong (in comparison to maximum amplitude on the trace) direct P phases which implies that the source released enough energy in the form of Pwave . These considerations led to a selection  of 170 records from 84 microseismic events. The dominant frequency bandwidth of the selected traces is 10-150 Hz.
* * *
Line 192: *"Figure 4"*

C: Can you also please comment on the frequency bandwidth of the passive data being used during the migration?

Thanks for the comment. This information was missing in the text and it's now included:
Line 171:
->The dominant frequency bandwidth of the selected traces is 10-150 Hz.
* * *
Line 196: *"Finally 10 CRGs"*

C: What is the specific reason for migrating in the common receiver domain rather than common shot domain? This is significantly reducing the already limited data. As was stated earlier that the event were located within an error of 5 ms. And traces were selected based on higher S/N ratio, therefore common shot gathers appears to be a more natural choice.

R: The reason is explained in details in this section (section 4.3) specifically in lines 181-191. Due to the specific characteristic of the dataset (the differences in the frequency of the source signal recorded at different stations), in our case it was more relevant to migrate common receiver gathers.
* * *
Line 204: *"ghost reflectors"*

C: Please specify what are ghost reflectors?

R: In seismic literature, "ghost reflector" is a known phrase. A short explanation is added:
Line 205:
(e.g. artificial reflectors due to reverberations)
* * *
Line 206: *"exponent value of 3"*

C: A single slice showing difference for alpha=1 and alpha=3 might bring a good way to showcase and validate the argument.

R: In order to avoid too many figures in the article, we excluded such a details. This is a simple exponential function which magnifies the differences between the most coherent values and the less coherent ones (please note that Cs(m,r)<1). We added here a short explanation to make the argument more clear. Because of the large data volume of the results of the migration procedure, it was not favourable to save all the results. On the other hand performing the whole procedure on the dataset would need time and computation facilities. We hope that the explanation would be sufficient.
* * *
Line 211: *"velocity model"*

C: As velocity model building is not the main highlight of the article, but still putting a few vertical sections in the vicinity of investigation area is a good idea. Another slide with traced rays superimposed on the same vertical slides will provide more confidence to the reader about which areas in the image domain are more likely to be correct. We can also have an opinion on the deeper reflectors as well.
Another reason to investigate the velocity model more here is because the semblance coefficient is calculated based on traveltimes ts and tr, therefore an error in velocity model might significantly impact the migrated image.

R: Figure 5 is added showing the velocity model. In this study we didn't perform ray tracing and therefore it is not illustrated. In line 354 it is described that the reliability of the velocity model is confirmed through a previous study.

->Figure 5 is added
* * *
Figure 7 (formerly figure 6):

C: Please change the depth scale to an equidistant scale [0,1,2,3,4,..]. It is particularly problematic during comparison between Fig. 6 & 7.

R: The depth scaling is changed on the figure:
->Figure7 is modified
* * *
Line 227: *"Figure 7"*

C: Does introducing an amplitude scale for the images will be beneficial here?

R: Because of the strong S-wave amplitudes as well as the amplitudes due to reverberations and ambient noise, scaling the amplitudes doesn't bold the reflections.
* * *
Line 236: *"coherency image shows the clearest and most pronounced image"*

C: I am not sure if I follow the exact explanation. As coherency image is actually depicting the weighting value (based on semblance coefficient) that should be applied at each image point in the Kirchhoff integral. It is not the migrated image? Did I get it right?

If yes, then it means that the reflectors are not at their true position yet because migration has yet not been performed. Please follow the red arrows I marked in Figure 6 (bottom left image). The reflectors marked by red arrow are prominently present in the phase-consistent image which also corresponds well with the coherency image. But the deeper reflectors present in the coherency image are not so well pronounced in the other two image cubes. This means a higher semblance coefficient (hence coherency image) does not certainly corresponds to a geological feature.

R: The coherency image is in fact the migrated image. Referring to the equation 2, the coherency image is obtained by keeping the other fields rather than $C_S(m, r)$ equal to 1. Thus in the final migrated image, an amplitude equal to the coherency value is assigned to each image point regardless to the recorded amplitude of the wavefield.
* * *
Line 256: "all sources have the same contribution in the coherency image irrespective of their P-wave amplitude"

C: Please elaborate how!

R: The explanation is added:

Line 256-257:
-> referring back to the equation [3], the effect of the amplitude variety of different sources on the calculated coherency values is very low. In other words, all sources have almost
* * *
Line 271: *"bending point and this may explain the sudden change"*

C: I will avoid such an interpretation here because the Schwerin fault was not at all been mapped. Therefore it is not most certain that the bending of the Rotter Kamm fault is due to Schwerin fault.

R: Based on the geological information of the area, the approximate extension of the Schwerin fault is presented in the geology map of the area (see Hiller and Schuppan, 2008). In our interpretations, we explain that comparing our imaging results and the geological information of the area, that the Schwerin fault **may be** the cause of the bending in the extension of the Roter Kamm (rather a hypothesis) and we don't claim this as a fact at all. Please see lines 271-274.

Line 275: *"375"*

C: I think you mean 125 m.

R: Actually we mean a total interval of 375 m (7625m to 8000m).
* * *
Line 276: *"Figure 11"*

C: As coherency migration is an advanced version of prestack KPSDM, adding few comparison slices of KPSDM along side of coherency image, and phase-consistent or absolute value is necessary in order to compare & validate the novelty of coherency migration.

R: In figure 7 we compare the coherency image with the phase-consistent and the absolute values image. Comparing the coherency image directly with the two other images shows also the functionality of the coherency migration in comparison to KPSDM, i.e. when the coherency values would be excluded from the two other images. It is clear that in this case, the phase-consistent and absolute values images will present only smeared amplitudes (along TWT migration isochrones). In other words, the result of KPSDM will be the same as the phase-consistent and absolute values images, but without the pronounced reflectors shown in figure 8.
* * *
Line 293: *"our final image"*

C: our final image (coherency image).

R: The line is modified:
Line 293:
->(coherency image)
* * *
Line 295: *"active seismic image cube"*

C: Please specify- is it PreStack, Post-stack or CRS image?

R: The explanation is added. ( "result of coherency migration" = PreStack. Earlier in section 3 it is described in detail that the coherency migration is a PreStack migration procedure):
Line 295:
-> which is also resulted from coherency migration
* * *
Line 315: *"Figure 12"*

C: Luschen et. al had thoroughly presented the 3D-CRS & 3D-FD post-stack time migration, and a pre-stack time migration based on Kirchhoff migration was also calculated. If it is possible, kindly provide a

comparison slide of time domain result as well for a more appropriate comparison. I assume that here the right hand side figure is coherency image calculated using the final time-domain processed data as input.

R: The right hand side figure (Fig.12) is the result of coherency migration in depth-domain (Kirchhoff prestack depth migration with the additional coherency weighting factor). In our imaging procedure we didn't perform migration in time-domain and therefore no comparison in time domain is presented.
* * *
C: Please put in one or two sentences about the computational time and resources required for KPSDM and CM!

R: The explanation is added to lines 207-208:
Line 207-208:
-> We performed the Coherency Migration over a migration cube (11.5 x 14 x 9 km) with 25 m spacing between grid points. Using 8 double core 2.3 GHz processors in a parallel computation procedure, the computations required about 5 hours.
* * *
Line 325: *"the phase consistent active"*

C: The reflectors on the left are contracted laterally in here, something which we expect after the migration. As I commented earlier, if the coherency image is representing only the weighting values based on semblance coefficient, then the interpretation should rather be based on phase-consistent or absolute value migrated results.

R: Please note that the imaging procedure is done using very low number of traces. On the other hand, the sources are passive sources with a wide range of source wavefield frequency and variable duration of the source energy release which makes the imaging procedure more complex in comparison to the conventional seismic survey. Therefor evaluating the expected results from imaging phase-consistent or absolute values image (as in the conventional active seismic imaging with tens of thousands recorded traces) won't be necessarily more convenient. Referring back to the figure 7 and also analysing the 3 images (discussed in lines 236-260), it is shown that imaging the calculated coherency values presents the most clear and reliable image. This is proved by comparing this image to the results of the active seismic survey in figures 12 and 13.

In section 3 we described that the coherency values represents the semblance of the recorded wavefields which consequently can represent the common origin of the recorded amplitudes (i.e. refraction points in depth). Therefore, more than being used as a weighting factor, this value can also be used as an image value itself.

Line 168: " we selected only those records with clear and strong (in comparison to maximum amplitude on the trace)"

C: "direct P phases. " / It is much more effective if the authors specified the uncertainty level in P phase picking.

R: The microseismic events used in this study are located through a migration-based event location approach. The localization procedure including the uncertainty level of P-phase picking (3 ms) has been discussed in a previous publication (Hassani et al. 2018) which is referenced in line 164.

Line 188: "As can be seen, the wavefield for the same event appears with"

C: "different frequency content at different stations.". / It is better that the event specification (time, magnitude, place) will be included.

R: The hypocenter location of the events are illustrated in figures 1, 2 and specifically in figure 6. However since the global coordinate of the located hypocenters is not directly relevant to the theme of this study, we avoided to go through those details. This applies also to the specific time of the events. The magnitude range of all events is mentioned in line 159 (−1.30<Mw<0.90).